# Potential Application of Marine Algae and Their Bioactive Metabolites in Brain Disease Treatment: Pharmacognosy and Pharmacology Insights for Therapeutic Advances

**DOI:** 10.3390/brainsci13121686

**Published:** 2023-12-07

**Authors:** Miski Aghnia Khairinisa, Irma Rahayu Latarissa, Nadiyah Salma Athaya, Vandie Charlie, Hanif Azhar Musyaffa, Eka Sunarwidhi Prasedya, Irma Melyani Puspitasari

**Affiliations:** 1Department of Pharmacology and Clinical Pharmacy, Faculty of Pharmacy, Padjadjaran University, Sumedang 45363, Indonesia; irmarahayulatarissa@gmail.com (I.R.L.); nadiyah19002@mail.unpad.ac.id (N.S.A.); vandie19001@mail.unpad.ac.id (V.C.); hanif19001@mail.unpad.ac.id (H.A.M.); irma.melyani@unpad.ac.id (I.M.P.); 2Centre of Excellence in Pharmaceutical Care Innovation, Padjadjaran University, Sumedang 45363, Indonesia; 3Department of Biology, Faculty of Mathematics and Natural Sciences, University of Mataram, Mataram 83115, Indonesia; ekasprasedya@unram.ac.id; 4Bioscience and Biotechnology Research Centre, Faculty of Mathematics and Natural Sciences, University of Mataram, Mataram 83126, Indonesia

**Keywords:** marine, algae, brain health, neurology

## Abstract

Seaweeds, also known as edible marine algae, are an abundant source of phytosterols, carotenoids, and polysaccharides, among other bioactive substances. Studies conducted in the past few decades have demonstrated that substances derived from seaweed may be able to pass through the blood–brain barrier and act as neuroprotectants. According to preliminary clinical research, seaweed may also help prevent or lessen the symptoms of cerebrovascular illnesses by reducing mental fatigue, preventing endothelial damage to the vascular wall of brain vessels, and regulating internal pressure. They have the ability to control neurotransmitter levels, lessen neuroinflammation, lessen oxidative stress, and prevent the development of amyloid plaques. This review aims to understand the application potential of marine algae and their influence on brain development, highlighting the nutritional value of this “superfood” and providing current knowledge on the molecular mechanisms in the brain associated with their dietary introduction.

## 1. Introduction

Since the brain processes, integrates, and coordinates information from peripheral sense organs and responds by centrally elaborating instructions appropriately conveyed back to the peripheral in each body area, it serves as the body’s central control center for most physiological activities. A few hormones control how the brain and neuronal cells grow and change in shape. Thyroid hormones (THs) impact the cerebellum’s granule cell migration and proliferation, cerebellar Purkinje cell dendritic growth, and cerebellar neuron synaptogenesis [1]. Consequently, the insufficiency of THs during development leads to disrupted motor coordination in adulthood [1,2]. In the hippocampus, hypothyroidism also disrupts the migration of granule cells and the dendritic growth of pyramidal cells [3,4], thus inducing aberrant synaptic function and learning [5].

Diet has an impact on how the brain develops, especially when it comes to certain molecules that are necessary for brain function. Maintaining synaptic plasticity and neuronal functions can be aided by appropriate dietary factors. A diet high in omega 3, choline, magnesium, B vitamins, vitamin D, certain amino acids, and phytoderivates (plant or seaweed-derived compounds) can, for example, preserve mental functions and support brain health while delaying the onset of mental and neurodegenerative diseases [6,7].

Numerous macro- and micronutrients, including minerals, amino acids, and vitamin B, as well as phytochemical compounds important to brain function, are found in seaweeds. When incorporated into the diet, they initially interact with the gastrointestinal tract’s microbiome [8,9]. Small bioactive molecules can be produced due to the interaction between seaweed and microbiota. The growth-promoting (prebiotic) effects of particular bacterial genera in producing neurotransmitters like serotonin and gamma-aminobutyric acid (GABA) affect intestinal ecology and host brain health [6,10]. Experimental evidence has shown that bioactive seaweed derivatives can reach and enter the brain and modulate multiple neuronal functions either directly through specific neuronal molecules and antioxidant and anti-inflammatory activities or indirectly through epigenetic mechanisms affecting the transcription of proteins involved in neurotransmissions, neuronal survival, and plasticity [7,8,9,11,12].

Marine algae have many bioactive chemicals that show health benefits. Algae metabolite compounds such as alkaloids, phenolics, terpenoids, phytosterols, carotenoids, and polysaccharides have been proven to have neuroprotective effects in preclinical models of neurodegenerative diseases. This compound has anti-inflammatory, antioxidant, and immunomodulatory activity associated with improving neurodegenerative diseases [11,13,14,15,16,17,18]. We review the potential of marine algae and their bioactive compounds for treating brain diseases through pharmacognosy and pharmacology approaches.

## 2. Seaweed Chemical Compound and Isolation

Bioactive substances, including polyphenols, carotenoids, vitamins, phycocyanins, phycobilins, and polysaccharides, abound in seaweeds. Many of these substances have positive uses for human health. Because of the polysaccharides on their cell surface, which allow them to retain inorganic marine substances, seaweeds also have an unparalleled richness of minerals and trace elements. Several of these essential minerals are found in seaweeds at relatively higher levels than in terrestrial food sources [19,20].

### 2.1. Brown Seaweeds

The brown seaweed’s cell wall contains anionic polysaccharides called alginic acid or alginate. These are insoluble and consist of alternating blocks of 1,4-linked polymer of β-D-mannuronic acid and α-L-guluronic acid. Sodium, potassium, or ammonium salts are alginates. By contrast, these fibers’ amorphous, slimy fraction consists mainly of water-soluble alginates and/or fucoidan [21]. Brown seaweeds are a source of nutrients different from those obtained from terrestrial plants and bestow many health benefits and bioprospecting potential [22]. The specific chemical compounds in brown seaweeds are listed in Table 1.

In one study [37], *Saccharina japonica* whole-plant powder was refluxed with methanol for 3 h. The total filtrate was then concentrated to dryness in a vacuum at 40 °C to produce methanol (MeOH) extract, which was suspended in distilled water and then partitioned successively with dichloromethane (CH_2_Cl_2_), ethyl acetate (EtOAc), and n-butanol (n-BuOH). The CH_2_Cl_2_ fraction was chromatographed on a silica-gel column using n-hexane: EtOAc (20:1) to obtain 18 subfractions. The best fraction was then subjected to column chromatography with a sequential elution of ethanol, MeOH, and acetone to obtain three subfractions. Repeated column chromatography was performed for the third subfraction with a solvent mixture of n-hexane and acetone (n-hexane: acetone, gradient 10:1–0:1) to isolate pheophorbide. Repeated chromatography was conducted on fraction 10 of a silica gel column to yield fucoxanthin, whose purity was checked by HPLC (P99%) [23].

*Sargassum swartzii* (*S. swartzii*) is one of the species of brown seaweed. In one study, the chemical compound sterols and oxysterols from *S. swartzii* were isolated first by rinsing the seaweed with water to remove any dirt, epiphytes, or impurities and then drying them at room temperature (25 °C ± 5 °C). The *S. swartzii* dried seaweeds were chopped, extracted using methanol, macerated, and stirred at room temperature. The soaking process was carried out for 24 h, after which the residue was filtered using the Whatman filter paper no. 1 filter. The *S. swartzii* filtrate was collected, concentrated under low pressure using a rotary evaporator, and dried under vacuum to obtain a dry methanol extract. The extract was then dissolved in a NaCl solution to the desired concentration before use [38].

### 2.2. Red Seaweeds

Red seaweeds have a variable protein content ranging from 10% to 50% of their dry weight; this value is higher than the protein content of the macroalgae group and some foods [39]. Similar to legumes, red seaweeds contain essential amino acids at about 25–50% [40]. Red seaweeds contain the greatest number of phenolic compounds, such as flavonoids, phenolic acids, and bromophenols; these components have different medical applications due to their reactions with proteins, such as enzymes or cellular receptors [41]. The specific compounds of red seaweeds are listed in Table 2.

Most of the cultivated red seaweed agar comes from the *Gracilaria* genus, but the *Gelidium* genus produces agar that is of a better quality than *Gracilaria*. Unfortunately, *Gelidium* is too overexploited, so its cultivation has been limited [54]. A previous study extracted *Gelidium amansii* by pouring 95% ethanol into a conical flask containing seaweed powder and solvent in a ratio of 50:1 (*v*/*w*) [43]. The mixture was stirred for 24 h in the dark on an orbital shaker (200 rpm at room temperature). After centrifuging the shaken slurry at 10,000 rpm, sterile cotton was used to filter the supernatant. A nitrogen gas stream was used to dry the filtrate after it had been concentrated using a vacuum. After weighing the dry ethanol extract, the yield extract (*w*/*w*%) was computed [43].

### 2.3. Green Seaweed

The majority of green seaweeds are regarded as a food source. Green seaweeds contain compounds such as taurine and lectins, fibers (ulvan sp, vitamins like tocol, etc.), and antioxidants like carotenoids, chlorophyll, bromophenol, and phloroglucinol. Green seaweeds also contain phenols, terpenoids, flavonoids, and amino acids like mycosporine (MAA) [55]. Other compounds found in specific green seaweeds are listed in Table 3.

*Caulerpa* sp. is one species of green seaweed with a high phenolic compound content. In one study, before extraction, fresh seaweed was cleaned of impurities that were still attached to it and then weighed [53]. The samples were divided into two, namely, fresh and boiled *Caulerpa* sp. For the boiled *Caulerpa* sp., the boiling temperature and time were 90 °C within 5 min. The sample was boiled using mineral water with a ratio of sample and water 1:4 (*w*/*v*). The water was heated first until it reached a temperature of 90 °C, then *Caulerpa* sp. was immersed in the boiling water for 5 min. The next step was a proximate, phytochemical, and total phenolic compound analysis. For the fresh *Caulerpa* sp., the sample was mashed first with a mortar, weighed, placed in an Erlenmeyer flask, and added to MetOH in a ratio of 1:2. The sample was soaked in methanol and then macerated using an orbital shaker for 24 h. The filtrate was then evaporated using a rotary vacuum evaporator at 40 °C. Extraction was carried out three times. The next step was to analyze the total phenolic compounds [58].

## 3. Mechanism and Effect on the Brain

The reactive oxygen species (ROS) or free radicals produced by the body can cause chronic human diseases, such as cancer, inflammation, and neurological disorders (including Alzheimer’s disease [AD], Parkinson’s, schizophrenia, and depression). The antioxidants in seaweeds improve drug health by reducing the free radicals in the body without damaging the body’s organs. Several compounds found in all types of seaweeds can work as antioxidants, such as phenolic compounds and flavonoids [68]. Some of the compounds from seaweeds, including fucoidan from *Sargassum fusiforme* and *Sargassum muticum* (brown seaweeds), phycoerythrin from *Porphyra* sp., *Gracilaria gracilis* (red seaweeds), and loliolide from *Codium tomentosum* (green seaweeds), have been tested and have shown effectiveness as antioxidants through various mechanisms. The neurotherapeutic potential effect of the marine-algae compounds is shown in Figure 1.

### 3.1. Phenolic Compounds

Phenolic compounds extracted from seaweed have been identified as possessing a neuroprotective effect [69] because they have very strong antioxidant properties [70]. Phenolic antioxidants hold significant promise in effectively neutralizing free radicals, significantly contributing to neuronal damage. As a result, they can have substantial neuroprotective effects and play a crucial role in managing neurodegenerative diseases [70].

In addition, phenolic compounds can also inhibit the enzymes acetylcholinesterase (AChE) and butylcholinesterase (BChE) and inhibit Aβ aggregation [69]. They can affect neurodegenerative diseases such as Alzheimer’s disease. The development of AD is associated with the disruption of the cholinergic pathway caused by the upregulation of acetylcholinesterase (AChE) and butylcholinesterase (BChE) [71]. Several studies have proven the reaction mechanism of phenols as antioxidants.

#### 3.1.1. Hydrogen Atomic Transfer

Hydrogen atom transfer occurs by transferring the hydrogen atoms in the antioxidants, which is phenol in this case, to the free radicals. Antioxidants containing H atoms are represented by (AH), which react and give H atoms to free radicals. Consequently, the free radicals turn into neutral forms, and the antioxidants are converted into free-radical antioxidants (A*) [72]. The mechanism of hydrogen atomic transfer is in Figure 2.

#### 3.1.2. Single Electron Transfer

Free radicals with energetically stable odd anions transfer their anions to antioxidants that are cation radicals so that the odd number of electrons formed in the antioxidants are distributed in the aromatic ring and throughout the molecule. The ionization potential becomes an obstacle in this mechanism because this reaction can only run when the free-radical ionization potential value is smaller than the antioxidant ionization potential, allowing the electrons in the free radicals to move to the phenolic compounds [73]. The mechanism of single electron transfer is shown in Figure 3.

#### 3.1.3. Sequential Proton Loss Electron Transfer

This mechanism begins with releasing protons from phenolic compounds into phenolic free radicals and anionic compounds according to their proton ability (PA). The PA value indicates the difficulty level of hydroxyl phenolic compounds to be dephosphorylated. The electrons or anionic compounds from the antioxidants then donate electrons to the free radicals. The mechanism of sequential proton loss electron transfer is shown in Figure 4.

#### 3.1.4. Transition-Metal Chelation

Transition-metal chelation (TMC) is a mechanism of antioxidants to chelate transition-metal compounds to form a stable metal. Phenolic compounds are good chelators; the chelation of Fe^3+^ metal directly reduces the formation of reactive OH free radicals from the Fenton reaction. As metal chelators with the ability to cross the blood–brain barrier (BBB), polyphenols can be used to treat neurodegenerative diseases [74].

### 3.2. Flavonoids

Many pharmacological effects are related to flavonoid antioxidants; their biological function is maintaining oxidative stress levels below the critical point [75]. The activity of flavonoids as antioxidants combines several pathways for reducing oxidase enzymes, such as cyclooxygenase, lipoxygenase, xanthine oxidase, myeloperoxidase, and NADPH oxidase. Flavonoids act as free-radical scavengers because of their structural stability, which can weaken highly reactive free radicals, so they turn into less reactive aroxyl radicals. Similar to phenol, the weakening mechanism of this oxidant is by donating electrons from free hydroxyl to free radicals so that free radicals can be neutralized [76]. The antioxidant capacity of flavonoids depends on the hydroxyl group position, sugar, double bonds, and molecule polarity [77]. Furthermore, in a model involving brain endothelial cells, it was found that certain flavonoids and their metabolites penetrated the blood–brain barrier and localized in the brain, indicating their significance as potential candidates for direct neuroprotective effects [78,79].

Flavonoids also affect cognitive effects through the cAMP response element-binding protein (CREB) pathway. They increase CREB phosphorylation in the hippocampus. CREB forms specific neuronal ensembles that encode new memories through cellular stimulation [80]. Among the many excitatory pathways to CREB, the majority of the cognitive benefits of flavonoids originate in the MEK → ERK → RSK/MSK → CREB → BDNF pathway [81]. The reaction mechanism of flavonoids as an antioxidant is shown in Figure 5.

### 3.3. Fucoidan

One of the causes of neurodegenerative disorders is the induction of lipopolysaccharide (LPS) and beta-amyloid (Aβ). Consequently, the body produces proinflammatory cytokines such as IL-1, TNF, PGE2, and NO through the MAP kinase pathway and nuclear factor (NF-kβ). Fucoidan blocks the Ikβ degradation and microglia activation stimulated by LPS and Aβ, so the above inflammatory factors cannot be produced [82]. It also acts as an anti-inflammatory by inhibiting the NF-Kβ and MAPK signaling pathways to suppress neuroinflammation and neurodegeneration [83,84]. In addition, fucoidan can penetrate the blood–brain barrier by binding to P-selectin on BBB endothelial cells. This, in turn, prevents the infiltration of leucocytes into the brain and reduces the inflammatory response [83]. The reaction mechanism of fucoidan as an inhibitor of microglia activation and NF-kβ is shown in Figure 6.

As an antioxidant, fucoidan can reduce the production of ROS stimulated by Aβ, LPS, and MPP^+^ and minimize oxidative stress in the brain [85]. The enzymes useful as natural antioxidants are superoxide dismutation (SOD) and glutathione peroxidase (GSH-Px). Fucoidan can increase the antioxidant capacity of PC12 cells by increasing the activities of SOD and GSH-Px. It can also decrease the malondialdehyde (MDA) level, an indicator of damage due to oxidative stress, by releasing LDH and reducing the concentration of ROS in PC12 cells [86]. Another study suggested that fucoidan can reduce tGCI-induced oxidative stress in the CA1 area in the hippocampus by reducing lipid peroxidation and increasing SOD expression [84,87].

### 3.4. Phycoerythrin

The antioxidant activity of phycoerythrin was tested by 2,2-azinobis-3-ethylbenzothiazoline-6-sulfonic acid (ABTS) and ferric-reducing antioxidant power (FRAP) test methods. ABTS is a reactive free radical, so the ABTS test determines the strength of antioxidant compounds to reduce or neutralize free radicals. The FRAP test also examined the antioxidant’s power by looking at its ability to reduce Fe^3+^ to Fe^2+^ [72].

The mechanism of phycoerythrin as an antioxidant has yet to be determined, but several studies have examined purified phycoerythrin by ABTS and FRAP tests and achieved good results, such as [88].A study showed that the phycoerythrin purified from *Pyropia yezoensis* has an antioxidant effect that can compete with glutathione as an antioxidant. According to [72], another study also reported that the purified phycoerythrin from *Spirulina platensis* showed DPPH inhibitory activity of up to 95.3% at a 200 g/mL dose. According to these two studies, phycoerythrin has a dose-dependent antioxidant effect. Therefore, this compound can protect the human body from the damage caused by ROS, such as attacking macromolecules and DNA, cancer, and neurodegenerative disorders [88].

### 3.5. Loliolide

Loliolide is a newly discovered metabolite in *Codium tomentosum* and has promising pharmacological effects, particularly neuroprotection. Silva et al. (2021) showed that loliolide works through various pathways, such as protecting nerve cells from the damaging effects caused by 6-OHDA, preventing mitochondrial dysfunction and apoptosis, and exhibiting anti-inflammatory effects by inhibiting the NF-kβ signaling pathway to reduce proinflammatory cytokines such as TNF- and IL-6. Given its mechanism and ability to cross the blood–brain barrier [89], loliolide might be a treatment for Parkinson’s disease. It can reduce cell death by protecting cells against damage from 6-OHDA, reducing oxidative stress, and blocking inflammatory pathways. As a new compound, further studies on the antioxidant, anti-inflammatory, and neuroprotective effects of loliolide are warranted [61]. The hypothesized mechanism of loliolide in 6-PLWHA that stimulates cell death is shown in Figure 7.

## 4. Pharmacological Potentials of Marine Algae: Evidence from In Vitro and In Vivo Studies

Several seaweeds have been tested for their activities related to brain health and have shown promising results for several diseases. The types of seaweed and the diseases that have been tested are listed in Table 4.

Oxidative stress is a substantial factor in AD’s clinical onset and advancement [98]. Several seaweeds have shown antioxidant properties that could potentially affect AD. *Saccharina japonica* reduced serum Aβ levels by 32.0% compared to the control group [90]. Aβ induces oxidative stress in neurons, resulting in elevated levels of hydrogen peroxide and lipid peroxides [99]. Similarly, *Undaria pinnatifida* can reduce apoptosis triggered by Aβ1-42 and increase neurite outgrowth activity at elevated concentrations, thereby displaying neuroprotective effects [80]. Fucosterol and two additional sterols, 3,6,17-trihydroxy-stigmasta-4,7,24(28)-triene and 14,15,18,20-diepoxyturbinarin, all of which were extracted from *Pelvetia siliquosa*, exhibited protective effects against oxidative stress induced by carbon tetrachloride (CCl4) by increasing the levels of superoxide dismutase (SOD), catalase, and glutathione peroxidase in the rats exposed to CCl4 challenge [32]. Two phlorotannins, dieckol and phlorofucofuroeckol were isolated from *Eisenia arborea*; at 10 μg/mL, these substances showed strong inhibitory activity against butyrylcholinesterase, a new target for the treatment of AD. All the compounds showed strong antioxidative properties and inhibitory activities against acetylcholinesterase, butyrylcholinesterase, and tyrosinase [26]. Acetylcholinesterase inhibitors are an important therapeutic strategy for AD. High acetylcholinesterase activity is associated with cholinergic dysfunction and memory impairment in AD [100]. Antioxidant activity, which is beneficial for AD, has also been observed for *Caulerpa lentillifera, Codium tomentosum,* and *Ulva lactuca*. Similar findings were reported for *Gelidium amansii* and *Kappaphycus alvarezii*. The neurodegenerative effect is related to the antioxidant activity [99,100].

24 (R, S)-Saringosterol isolated from *S. fusiforme* can cross the blood–brain barrier and might exert modulatory effects on the central nervous system to cope with disorders such as AD [101]. *Ulva rigida* has elevated levels of magnesium, which is crucial for the proper operation of the central nervous system and assists in alleviating the symptoms associated with Parkinson’s disease and AD [102].

In addition to their potential benefits for AD, certain types of seaweed hold promise for managing Parkinson’s disease. *Alaria esculenta* contains molecules that can modify the folding of the α-synuclein protein and inhibit its transformation into an amyloid structure. This transformation of α-synuclein from its naturally unfolded state and α-helical tetrameric form to an amyloid structure is a critical factor in the development of Parkinson’s disease [103].

*S. swartzii* has antidepressant effects. Its methanolic extract contains significant bioactive compounds, specifically sulfated polysaccharides and fucoxanthines [104,105]. Fucoxanthines, derived from brown seaweeds, such as *S. swartzii*, can reduce the levels of inflammatory cytokines, such as TNFα, inducible nitric oxide synthase, and COX-2. These cytokines play a pivotal role in the development of depression [106].

## 5. Safety Aspects

It is crucial to emphasize that, despite the substantial potential advantages of therapies involving marine algae and their bioactive metabolites, further research is required to enhance their safety. Research on the safe use of these therapies is limited, and possible side effects need to be further evaluated. Initial studies indicate that marine algae are generally considered safe to consume in moderate amounts [107]. Nevertheless, it is worth mentioning that marine algae that are tainted with harmful elements like heavy metals could present health risks. Both dried and fresh marine algae can potentially harbor heavy metals like cadmium, mercury, arsenic, or lead, which varies according to their source habitat [108]. This is linked to a higher likelihood of developing diabetes, cardiovascular disease, and neurodegenerative conditions, along with an elevated occurrence of lung, skin, and bladder cancer [37,109].

Marine algae are also among the most abundant dietary sources of iodine, a trace element essential in small quantities by the body. The recommended dietary allowance (RDA) for iodine in adults is 150 micrograms per day, with a tolerable upper limit of 1100 micrograms [109]. Excessive iodine intake can decrease thyroid hormone production and cause hypothyroidism. Research has indicated that consistent consumption of seaweed (several servings per week) can lead to elevated levels of thyroid-stimulating hormone, considered a risk factor for hypothyroidism [110]. In general, excessive iodine intake does not typically result in health issues for most individuals who are in good health. However, certain populations may be more susceptible to the adverse effects of excessive iodine and should consider restricting their consumption of seaweed products. These groups comprise individuals with pre-existing thyroid conditions, such as hypothyroidism (underactive thyroid) or hyperthyroidism (excessive thyroid hormone production), school-aged children, and infants [111]. Moreover, it is necessary to conduct further research to assess possible interactions with other drugs that patients with brain diseases may use. Marine-algae therapy may sometimes affect drug metabolism and trigger unwanted side effects [112].

## 6. Challenges and Future Perspective

The need for seaweed biomass is rising due to the increasing uses of seaweed; this demand cannot be satisfied by simply gathering them from the wild. Finding a suitable nutrient source that can support the large-scale production of marine algae remains a major bottleneck despite the need for more reports on nutrient sources for land-based cultivation. Aquaculture techniques must be strengthened to produce the greatest amount of seaweed biomass, and strong management strategies must be considered.

The source of the extraction process, synthesis, molecular weight, the presence of coextracts, the makeup of sugars, the degree of branching, etc. all impact the biological actions of the compounds found in marine algae. The metabolites’ biological actions and potential applications in the pharmaceutical, cosmeceutical, and nutraceutical industries could be enhanced through chemical or enzymatic modification. Therefore, defining these influencing parameters is necessary to successfully apply marine-algae products for human and animal nutritional and health benefits.

Using marine-algae compounds in food-contact packaging requires a cost-effective and sustainable supply of premium metabolites with bioactive qualities that can extend food storage without compromising nutritional value or quality. Moreover, the metabolite must be free of harmful substances from any microbes that may reside on the algal surface to develop edible food-packaging materials. Therefore, rigorous protocols must be planned to isolate and purify its metabolites to guarantee quality and purity.

Furthermore, the low bioavailability and limited water solubility of natural products, their physicochemical instability, their rapid metabolism, and their ability to cross the blood–brain barrier are some challenges and limitations that may impact their clinical efficacy. Furthermore, the blood–brain barrier restricts the amount of naturally occurring substances that can enter the brain and travel to the site of action. This will limit the distribution to brain tissues and result in low bioavailability. Delivering natural products and their isolated compounds through nanotechnology and nanocarrier-based techniques may help improve therapeutic responses and enhance their effectiveness. The bioavailability of natural products and their compounds can be increased by adding nanoparticles to the delivery system.

## 7. Conclusions

The study of marine algae and their bioactive metabolites presents a promising avenue for developing novel therapeutics for brain diseases. The rich diversity of marine ecosystems and the multifaceted pharmacological actions of these compounds provide a robust foundation for the further exploration and development of marine algae in pharmacognosy and pharmacology. Research has demonstrated the neuroprotective potential of marine-algae-derived compounds. They can mitigate oxidative stress, reduce neuroinflammation, modulate neurotransmitter levels, and inhibit the formation of amyloid plaques, which are crucial factors in the development and progression of brain diseases. Understanding the pharmacological mechanisms underlying the effects of marine-algae-derived compounds on brain diseases is essential. These mechanisms include anti-inflammatory, antioxidant, antiapoptotic, and neurotrophic actions, which collectively contribute to the therapeutic potential of these compounds. 

## Figures and Tables

**Figure 1 brainsci-13-01686-f001:**
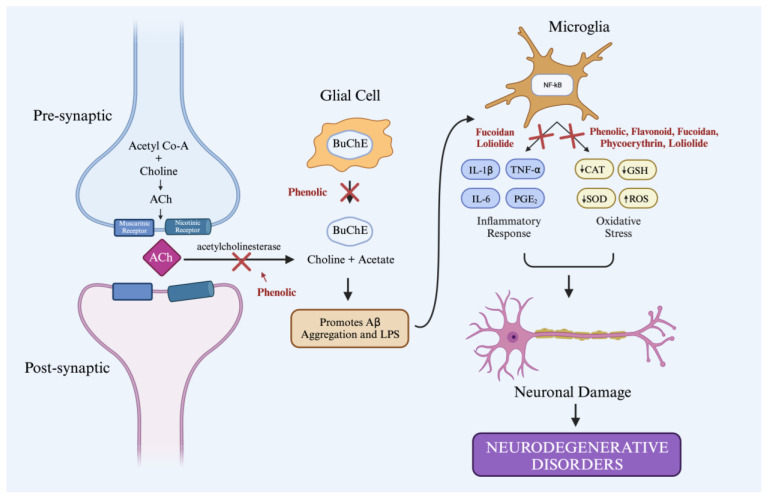
The neurotherapeutic potential effect of the marine-algae compounds. Created with Biorender.com, accessed on 1 October 2023.

**Figure 2 brainsci-13-01686-f002:**
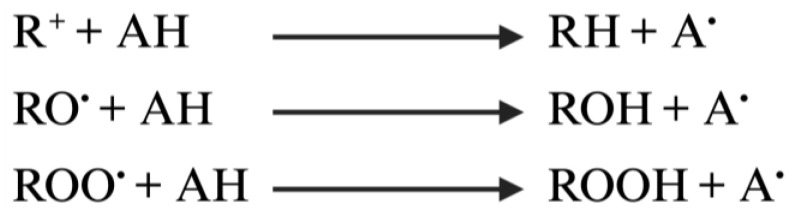
Mechanism of hydrogen atomic transfer.

**Figure 3 brainsci-13-01686-f003:**
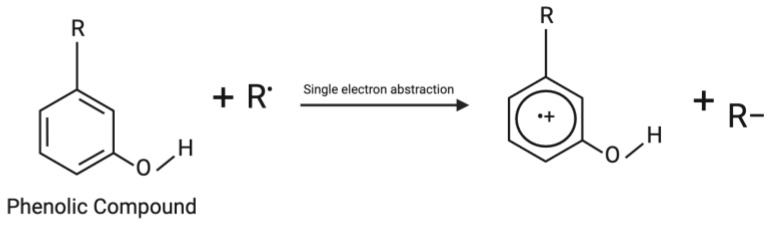
Mechanism of single electron transfer.

**Figure 4 brainsci-13-01686-f004:**
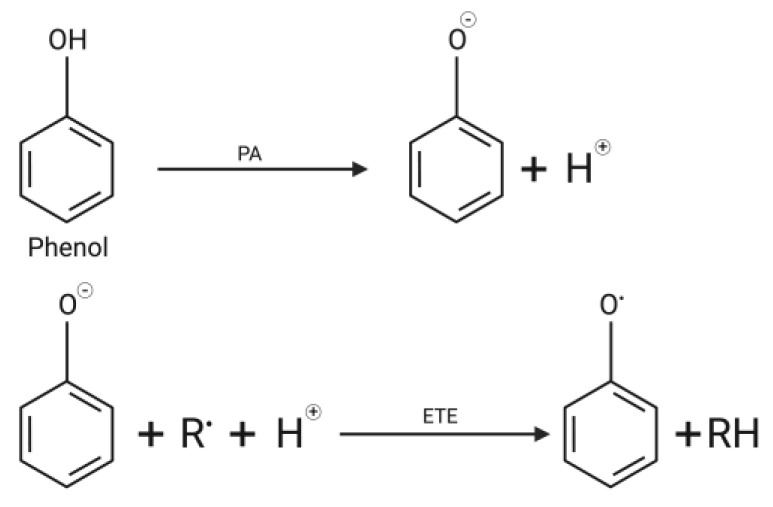
Mechanism of sequential proton loss electron transfer.

**Figure 5 brainsci-13-01686-f005:**
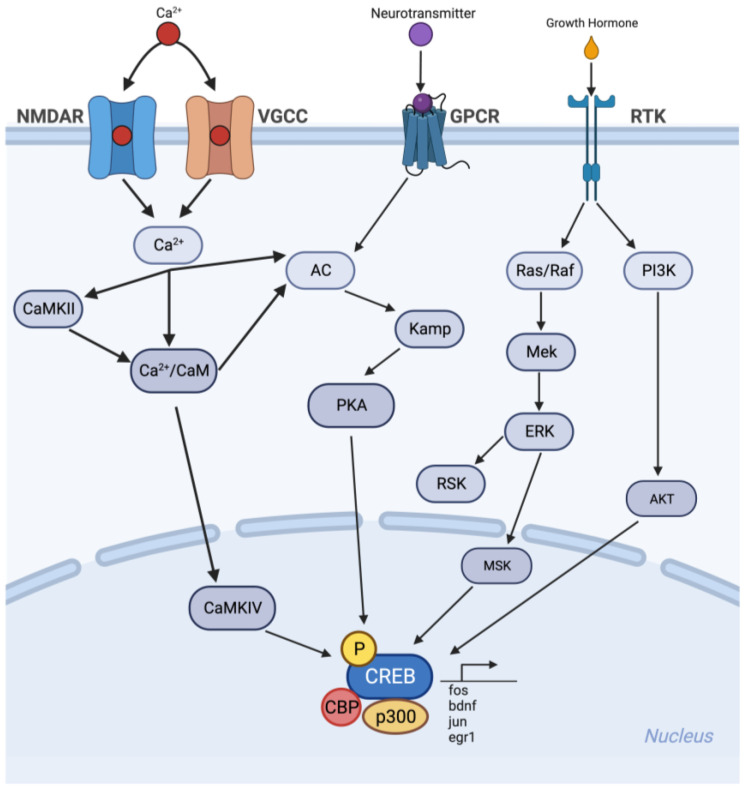
Reaction mechanism of flavonoid as an antioxidant. Created with BioRender.com, accessed on 1 October 2023.

**Figure 6 brainsci-13-01686-f006:**
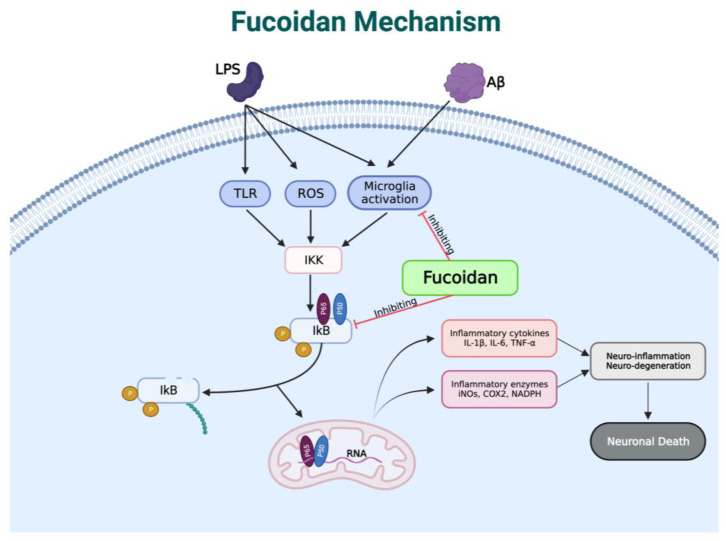
Reaction mechanism of fucoidan as an inhibitor of microglia activation and NF-kβ. Created with BioRender.com, accessed on 1 October 2023.

**Figure 7 brainsci-13-01686-f007:**
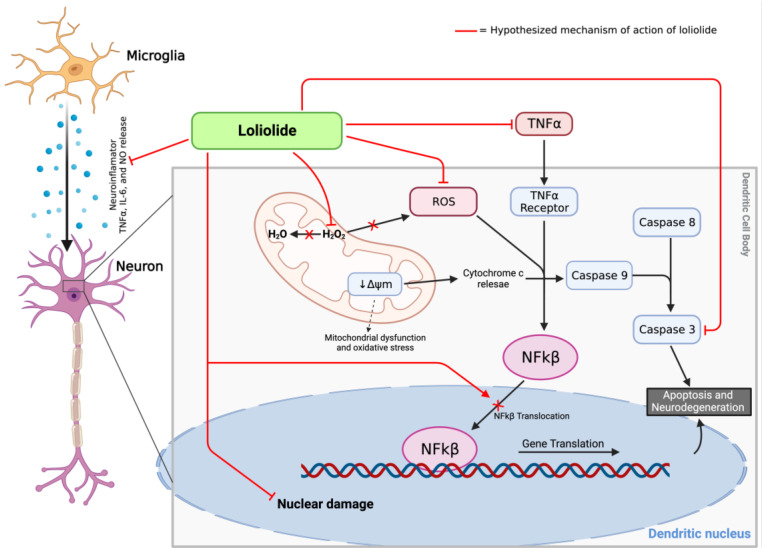
The hypothesized mechanism of loliolide in 6-PLWHA that stimulates cell death. Created with BioRender.com, accessed on 1 October 2023.

**Table 1 brainsci-13-01686-t001:** Specific chemical compounds of brown seaweeds.

Brown Seaweed Species	Chemical Compound						References
	Phenol	Polyphenol	Polysaccharide	Carotenoids	Flavonoids	Others	
*Saccharina japonica*	✓					γ-aminobutyric acid	[23]
*Undaria pinnatifid*		✓	✓		✓	Amino acid, Methacrylic acid	[24]
*Alaria esculenta*		✓					[25]
*Dictyota menstrualis*	✓						[26]
*Colpomenia sinuosa*	✓				✓	Steroid, Alkaloid	[27]
*Fucus spiralis*						Unsaturated fatty acid	[28]
*Ecklonia cava*		✓					[29]
*Pelvetia siliquosa*						Fucosterol	[30]
*Eisenia arborea*		✓					[31]
*Laminaria digitata*						Hexanal	[32]
*Sargassum fusiforme*	✓		✓		✓	alginic acid	[33]
*Sargassum muticum*	✓		✓	✓		Fatty acid, Terpenoid, Sterols	[34]
*Sargassum naozhouense*						carbohydrate, Protein	[35]
*Sargassum swartzii*	✓				✓	Terpenoid, Steroids, Coumarin, Sterols, Oxysterols	[36]

**Table 2 brainsci-13-01686-t002:** Specific Chemical Compounds in Red Seaweeds.

Red Seaweed Species	Chemical Compound						References
	Phenolic	Polyphenol	Polysaccharide	Carotenoids	Flavonoids	Others	
*Porphyra* sp.			✓			Iron, Vitamin B12, Phycoerythrin, Phycocyanin	[42]
*Gelidium amansii*						Glucose, Galactose, 3,6-AHG	[43]
*Gracilaria gracilis*				✓		Lipid, Sterol, Phycoerythrin, Phycocyanin	[44]
*Gracilaria verrucosa*	✓	✓			✓	Steroid, Saponin, Quinones	[45,46]
*Gracilariopsis chorda*						Arachidonic acid	[47]
*Hypnea musciformis*	✓					Carbohydrate, Protein, α-tocopherol, L-ascorbic acid	[48]
*Gloiopeltis tenax*	✓					Vanillylacetone, Fatty acids, Tetradecanoic acid, Linoleic acid, Oleic acid	[49]
*Eucheuma cottonii*					✓	Steroid, Alkaloid, Triterpenoid	[50]
*Kappaphycus alvarezii*	✓				✓	Alkaloid, Carbohydrate, Terpenoid	[51]
*Pyropia orbicularis*	✓			✓		Phycocyanin	[52]
*Ahnfeltia plicata*	✓				✓	Alkaloid, Amino acid, Tannin, Coumarins	[53]

**Table 3 brainsci-13-01686-t003:** Specific Chemical Compounds in Green Seaweeds.

Green Seaweed Species	Chemical Compound						References
	Phenolic	Polyphenol	Polysaccharide	Carotenoids	Flavonoids	Others	
*Caulerpa lentillifera*	✓				✓	siphonaxanthin	[56]
*Caulerpa racemosa*	✓				✓		[57,58]
*Enteromorpha prolifera*	✓	✓					[59]
*Enteromorpha clathrata*	✓						[59]
*Codium tomentosum*	✓	✓			✓	phlorotannins, loliolide	[60,61]
*Ulva pertusa*			✓			ulvan	[62]
*Codium fragile*	✓	✓			✓		[63]
*Ulva lactuca*	✓			✓		ulvan, starch, water-soluble cellulose	[64,65]
*Ulva prolifera*		✓	✓	✓		terpenoid, alkaloid, peptide, chlorophyll, sterol	[66]
*Ulva rigida*	✓						[67]

**Table 4 brainsci-13-01686-t004:** Pharmacological Potentials of Marine Algae.

Species	Division	Research for	Reference
*Saccharina japonica*	*Phaeophyta*	Epilepsy, Anxiety, Schizophrenia, Parkinson, Alzheimer	[90]
*Undaria pinnatifida*	*Phaeophyta*	Alzheimer	[91]
*Alaria esculenta*	*Phaeophyta*	Parkinson	[25]
*Pelvetia siliquosa*	*Phaeophyta*	Alzheimer	[92]
*Eisenia arborea*	*Phaeophyta*	Alzheimer	[31]
*Sargassum fusiforme*	*Phaeophyta*	Alzheimer	[93]
*Sargasum swartzii*	*Phaeophyta*	Depression	[38]
*Caulerpa lentillifera*	*Chlorophyta*	Alzheimer	[57]
*Codium tomentosum*	*Chlorophyta*	Alzheimer, Depression, Parkinson	[60]
*Ulva lactuca*	*Chlorophyta*	Alzheimer	[94]
*Ulva rigida*	*Chlorophyta*	Parkinson, Alzheimer, Antioxidant	[95]
*Gelidium amansii*	*Rhodophyta*	Neurodegenerative	[96]
*Kappaphycus alvarezii*	*Rhodophyta*	Neurodegenerative	[97]

## Data Availability

All data found and analyzed are included in this review article.

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
