# Peer review of "Potential Application of Marine Algae and Their Bioactive Metabolites in Brain Disease Treatment: Pharmacognosy and Pharmacology Insights for Therapeutic Advances"

_brainsci, 2023, doi:10.3390/brainsci13121686_

Round 1

Reviewer 1 Report

Comments and Suggestions for Authors

The current manuscript is an interesting review on the potential of marine algae and their bioactive metabolites as neurotherapeutics. Many relevant studies were included, and the analysis is thorough. Hence, I only ask that the following changes be made before acceptance for publication:

- The abstract should not include “examples”, but general conclusions; more specific conclusions of the review should be added, or all studied compounds/molecules/algae (not just 1 example);

- Permissions for the reproduction of the reused images should be mentioned in the respective captions;

- More should be said about the ability of each mentioned compound to cross the BBB: do they cross substantially, which is the transport mechanism (active, passive, etc.), do they undergo p-gp efflux, etc.;

- An image summarizing the neurotherapeutic potential of the mentioned compounds should be made and added;

- More should be said on the future studies necessary to put these compounds on the market as therapeutic options; additionally, how should they be administered, in what concerns administration route, formulation, etc.;

- Safety aspects of these compounds should also be mentioned.

Author Response

Dear Reviewer

Thank you very much for your feedback on this manuscript. I've attached a file to our response.

Best regards,

Author

Reviewer 2 Report

Comments and Suggestions for Authors

This article provides a brief review of the application and research of active ingredients in algae in degenerative encephalopathy. Overall, the paper lacks clear thinking, prominent focus, and it has not elaborated the research and mechanism of action of algal components on brain diseases in depth. It is not enough to use antioxidant and anti-inflammatory to illustrate the treatment of brain diseases. 

1.     The Introduction section does not link algae to brain diseases, and the content of each paragraph appears to be relatively scattered.

 2. “2. Edible seaweeds” What is the purpose of this section? Do you still need such a detailed introduction to seaweed? Not closely related to the topic of the paper.

 3. There are many errors in the section of “3. Seaweed Chemical Compound and Isolation”, e.g. in 3.1, the natrium salts of alginic acid is soluble; in Table 2, fucoxanthin is a type of carotenoids, tannin is a phenol compound, and fucoidan is a polysaccharide, etc.

 4. L109-130, L146-152, and L164-176 mainly cite separation and analysis methods from literature, which are of no use to this paper.

 5. 4.1, including 4.1.1, 4.1.2, 4.1.3, and 4.1.4, are all common sense content and do not mention the association between polyphenols and neurodegenerative diseases. How many scholars are currently conducting research on polyphenols and brain diseases, and what are the effectiveness and mechanism of the research?

 6. The content of L273-277 and L314-318 is not very relevant to the topic of the manuscript.

1.     The Introduction section does not link algae to brain diseases, and the content of each paragraph appears to be relatively scattered.

 2. “2. Edible seaweeds” What is the purpose of this section? Do you still need such a detailed introduction to seaweed? Not closely related to the topic of the paper.

 3. There are many errors in the section of “3. Seaweed Chemical Compound and Isolation”, e.g. in 3.1, the natrium salts of alginic acid is soluble; in Table 2, fucoxanthin is a type of carotenoids, tannin is a phenol compound, and fucoidan is a polysaccharide, etc.

 4. L109-130, L146-152, and L164-176 mainly cite separation and analysis methods from literature, which are of no use to this paper.

 5. 4.1, including 4.1.1, 4.1.2, 4.1.3, and 4.1.4, are all common sense content and do not mention the association between polyphenols and neurodegenerative diseases. How many scholars are currently conducting research on polyphenols and brain diseases, and what are the effectiveness and mechanism of the research?

 6. The content of L273-277 and L314-318 is not very relevant to the topic of the manuscript.

Comments on the Quality of English Language

None.

Author Response

(The authors gave the same response as above.)
